# Impacts of Calcium Addition on Humic Acid Fouling and the Related Mechanism in Ultrafiltration Process for Water Treatment

**DOI:** 10.3390/membranes12111033

**Published:** 2022-10-23

**Authors:** Hui Zou, Ying Long, Liguo Shen, Yiming He, Meijia Zhang, Hongjun Lin

**Affiliations:** 1College of Geography and Environmental Sciences, Zhejiang Normal University, Jinhua 321004, China; 2Department of Materials Science and Engineering, Zhejiang Normal University, Yingbin Road 688, Jinhua 321004, China

**Keywords:** membrane fouling, humic acid, calcium ion, hydraulic resistance, ultrafiltration process

## Abstract

Humic acid (HA) is a major natural organic pollutant widely coexisting with calcium ions (Ca^2+^) in natural water and wastewater bodies, and the coagulation–ultrafiltration process is the most typical solution for surface water treatment. However, little is known about the influences of Ca^2+^ on HA fouling in the ultrafiltration process. This study explored the roles of Ca^2+^ addition in HA fouling and the potential of Ca^2+^ addition for fouling mitigation in the coagulation-ultrafiltration process. It was found that the filtration flux of HA solution rose when Ca^2+^ concentration increased from 0 to 5.0 mM, corresponding to the reduction of the hydraulic filtration resistance. However, the proportion and contribution of each resistance component in the total hydraulic filtration resistance have different variation trends with Ca^2+^ concentration. An increase in Ca^2+^ addition (0 to 5.0 mM) weakened the role of internal blocking resistance (9.02% to 4.81%) and concentration polarization resistance (50.73% to 32.17%) in the total hydraulic resistance but enhanced membrane surface deposit resistance (33.93% to 44.32%). A series of characterizations and thermodynamic analyses consistently suggest that the enlarged particle size caused by the Ca^2+^ bridging effect was the main reason for the decreased filtration resistance of the HA solution. This work revealed the impacts of Ca^2+^ on HA fouling and demonstrated the feasibility to mitigate fouling by adding Ca^2+^ in the ultrafiltration process to treat HA pollutants.

## 1. Introduction

Due to its high efficiency in removing various pollutants, the ultrafiltration process has been widely applied to treat wastewater and surface water [1,2,3]. Nevertheless, the existence of natural organic matter (NOM) in natural water and wastewater bodies would cause serious membrane fouling and thus hinder the promotion of application of low-pressure membranes such as ultrafiltration membrane [4,5,6,7,8,9]. It is generally accepted that coagulation and flocculation can serve as a pretreatment step for ultrafiltration process as it can cluster foulant particles and absorb NOM, and therefore simultaneously reduce membrane fouling and improve NOM rejection [10,11,12].

NOM is a mixture of organic compounds that come from nature and composed of various substances such as humic acid (HA), protein, and polysaccharides [1,13,14,15]. Among them, HA is considered one of the most important categories that contribute to membrane fouling. It is ubiquitous in aquatic ecosystems, and the concentration distribution range varies from a few mg/L of dissolved organic carbon (DOC) to more than a few hundred mg/L DOC [16,17,18,19]. Extensive studies have reported the significant contribution of HA to membrane fouling [20,21,22,23]. Unlike protein and polysaccharides, HA has a relatively small molecular weight [20]. Therefore, it cannot be completely removed by the ultrafiltration process and also would cause more severe irreversible membrane fouling [24]. Previous literature has reported that coagulation is an effective pretreatment approach to mitigate membrane fouling caused by HA [21]. Nevertheless, the external addition of flocculant apparently would increase the maintenance cost. Therefore, a promising and cost-effective strategy is to make full use of the flocculating substances coexisting with HA in natural water and wastewater.

Since HA bears lots of functional groups (such as hydroxylm, ethoxy, and carboxyl) and binding sites, calcium ions (Ca^2+^) might be an excellent natural flocculant [25]. Ca^2+^ is a common metal ion in surface water and its concentration in municipal wastewater is reported to be in the range of 0.5–3 mM [26]. As a divalent cation, Ca^2+^ has a bridging effect and can promote biological flocculation in sewage, which is bound up with membrane fouling. Some studies have explored the roles of Ca^2+^ in the membrane fouling performance of HA in different membrane filtration processes [27,28,29,30,31]. A consistent result of the enhanced HA fouling caused by Ca^2+^ addition has been reported in the filtration processes of anion exchange, nanofiltration, and forward osmosis [29,32,33]. However, unlike the filtration processes mentioned above, the studies regarding the effects of Ca^2+^ on HA in the ultrafiltration process obtained contradictory results. Lin et al. [34] pointed out that the membrane fouling was improved due to the increased Ca^2+^ concentration. Wang et al. [31] pointed out that Ca^2+^ has a more effective capacity in ultrafiltration fouling intensification than Mg^2+^. Nevertheless, Li et al. [35] found that Ca^2+^ can promote the formation of reversible fouling and thus can achieve a higher removal efficiency of HA. The inconsistent results suggest that the effects of Ca^2+^ on membrane fouling are complex and require further study.

The causes of the inconsistent results in the previous studies may lie in several aspects. First, the effects of Ca^2+^ on HA-induced membrane fouling depend on the membrane material. However, the materials of the ultrafiltration membrane applied in the literature differed in different studies. In addition, previous studies evaluated the HA–Ca^2+^ fouling through a single or whole filtration resistance variation. For example, Lin et al. [34] investigated the effects of Ca^2+^ on HA fouling for the polyvinylchloride (PVC) membrane through the interfacial interaction energy change. Chang et al. [36] mainly focused on the HA–Ca^2+^ effects in hydraulically irreversible fouling. Previous studies did not well distinguish the different filtration resistance components. Furthermore, the effects of specific Ca^2+^ concentration on HA fouling depend on the HA concentration. In fact, studies were seldom with regard to the HA–Ca^2+^ fouling of polyvinylidene fluoride (PVDF) ultrafiltration membrane [37]. However, PVDF is one of the most widely used membrane materials in wastewater treatment [38,39]. Therefore, more studies are of great significance for HA fouling control in the PVDF ultrafiltration process.

Therefore, a simplified model of the separation membrane that functions in the cross-flow filtration mode was adopted. The effects of Ca^2+^ on HA fouling were evaluated through different hydraulic resistance components including concentration polarization, deposit, internal, and membrane fouling. The properties of the HA–Ca^2+^ complexes were analyzed by using a series of characterization methods. Finally, thermodynamic interaction theory was used to analyze the possible mechanisms.

## 2. Materials and Methods

### 2.1. Sample Preparation

All the reagents and chemicals applied in the current study were purchased from Sinopharm Chemical Reagent Co., Ltd. The experimental sample was prepared according to the following steps. First, 1 g of HA was dissolved in 1000 mL of NaOH (pH = 13) and continuously stirred for 24 h to make sure the complete dissolution of HA. Next, the pH of the stock solution was adjusted to 7.0 by using 1 mol/L HCl solution and then stored at room temperature. In the current work, a HA concentration of 100 mg/L was adopted to simulate the HA content in natural water, and the working solution was prepared by diluting the stock solution with deionized water. A specific volume of the stock CaCl_2_ solution was added during the dilution process to obtain the set Ca^2+^ concentration. It should be noted that the selected HA concentration (100 mg/L) is higher than that in the natural water body in order to facilitate the formation of membrane fouling. Similar concentration levels were typically used for lab-scale studies in the previous literature [40,41].

### 2.2. Filtration Resistance Tests

A lab-scale cross-flow filtration system (customized by Hangzhou Jiuling Technology Co., Ltd., Hangzhou, China) was applied for filtration resistance tests, and all the tests were conducted at room temperature with an operating pressure of 2 bar. The membrane utilized in this study was made of PVDF material (Shanghai SINAP Co., Ltd., Shanghai, China), and the effective membrane surface area was 25 cm^2^. The membrane was characterized as having a 0.1 µm pore size with a 140 kDa molecular weight cutoff (MWCO).

The filtration resistance was determined according to the Darcy–Poiseuille equation described as follows [42]:(1)$$J_{f}=\frac{1}{R_{m}+R_{e}+R_{p}+R_{i}}\frac{\Delta P}{\eta_{\mathrm{water}}}$$
where *J_f_* is the filtration flux; *R_m_*, *R_e_*, *R_p_*, and *R_i_* are the membrane filtration resistance, membrane surface deposit resistance, concentration polarization resistance, and internal blocking resistance, respectively; ∆*P* is the transmembrane pressure; and $\eta_{\mathrm{water}}$ is the dynamic viscosity of water. In the Poiseuille equation, the viscosity of the filtrate is equivalent to that of water.

The membrane filtration resistances were tested by filtering deionized water through the virgin membranes. Before the tests, the membranes were pre-compressed under 5 bar for at least 1 h to obtain a steady pure water flux. For each membrane, at least 3 tests were conducted to obtain an average value. The values of the membrane filtration resistance were calculated according to Equation (2):(2)$$R_{m}=\frac{1}{J_{f}}\frac{\Delta P}{\eta_{\mathrm{water}}}$$

By filtering the HA suspension, the total filtration resistances were estimated by Equation (3):(3)$$R_{T}=\frac{1}{J_{f}}\frac{\Delta P}{\eta_{\mathrm{water}}}=R_{m}+R_{e}+R_{p}+R_{i}$$

After the filtration of the HA suspension, the membranes were rinsed with deionized water three times to eliminate all traces of the solution, especially the concentration polarization layer. Thereafter, deionized water was filtered through the rinsed membrane to obtain resistance *R*_1_, which is the sum of *R_m_*, *R_e_*, and *R_i_*:(4)$$R_{1}=R_{m}+R_{e}+R_{i}$$

Afterward, the deposit formed on the membrane surface was removed by a sponge followed by ultrasonic wave treatment. *R*_2_, the sum of *R_m_* and *R_i_*, was then obtained by filtration of deionized water through the cleaned membrane:(5)$$R_{2}=R_{m}+R_{i}$$

Based on Equations (2)–(5), the values of *R_p_*, *R_e_*, and *R_i_* were estimated by Equations (6)–(8):(6)$$R_{p}=R_{T}-R_{1}$$
(7)$$R_{e}=R_{1}-R_{2}$$
(8)$$R_{i}=R_{T}-R_{m}-R_{e}-R_{p}$$

### 2.3. Analytical Methods

The functional groups of the samples were determined by a Fourier transform infrared spectrometer (FTIR, NEXUS 670, Waltham, MA, USA). The wavenumber range was 4000–500 cm^−1^. The particle size distribution (PSD) of the HA suspensions with different Ca^2+^ concentrations was measured by a particle size analyzer (Mastersizer 3000, Malvern, UK). Triplicate measurements were conducted for each sample. The total organic carbon (TOC) content of the HA solution was determined by a TOC analyzer (Liqui TOCII, Elementar, Hanau, Germany). The contact angle of the PVDF membrane and HA samples was determined by a contact angle meter (Kino Industry Co., Ltd., Boston, MA, USA), and the operation was similar to the previous reports. Zeta potential of the HA solutions and membrane surface was measured by a Malvern Zetasizer Nano ZS and a zeta 90 Plus instrument, respectively. Details regarding the operations of the abovementioned characterization can be found in the previous publications [43,44,45,46,47,48].

### 2.4. Extended Derjaguin–Landau–Verwey–Overbeek (XDLVO) Theory

It has been reported that the short-ranged thermodynamic interactions between foulants and membrane surface play a key role in the adhesion of different foulants on the membrane. The thermodynamic interactions can be divided into three parts according to the XDLVO theory [49,50,51], which are van der Waals (LW), acid–base (AB), and electrostatic double-layer (EL) interaction energies. The strength of these energies at separation distance (h) ($\Delta G^{LW}(h)$, $\Delta G^{EL}(h)$, and $\Delta G^{AB}(h)$) (mJ∙m^−2^) can be quantified by the following equations [52,53]:(9)$$\Delta G^{LW}(h)=\Delta G_{h_{0}}^{LW}\frac{h_{0}^{2}}{h^{2}}$$
(10)$$\Delta G^{EL}(h)=\varepsilon_{r}\varepsilon_{0}\kappa \zeta_{1}\zeta_{3}\left(\frac{\zeta_{1}^{2}\zeta_{3}^{2}}{2\zeta_{1}\zeta_{3}}\left(1-\coth\kappa h\right)+\frac{1}{\sinh\kappa h}\right)$$
(11)$$\Delta G^{AB}(h)=\Delta G_{h_{0}}^{AB}\exp\left(\frac{h_{0}-h}{\lambda }\right)$$
where *h* and *h*_0_ are the separation distance (nm) and minimum separation distance (nm) between two entities, respectively; *ε_r_ε*_0_ is the solution dielectric constant (C∙V^−1^∙m^−1^); *κ*, *ζ*, and *λ* represent the reciprocal of the Debye length (nm^−1^), surface zeta potential (mV), and the attenuation of AB interaction (usually assigned as 0.6), respectively; the subscripts 1, 2, and 3 mean the membrane, pure water, and foulant, respectively; $\Delta G_{h_{0}}^{LW}$, $\Delta G_{h_{0}}^{AB}$, and $\Delta G_{h_{0}}^{EL}$ are the interaction energies at a separation distance of *h*_0_ (mJ∙m^−2^), which can be quantified by Equations (12)–(14), respectively:(12)$$\Delta G_{h_{0}}^{LW}=-2\left(\sqrt{\gamma_{1}^{LW}}-\sqrt{\gamma_{2}^{LW}}\right)\left(\sqrt{\gamma_{3}^{LW}}-\sqrt{\gamma_{2}^{LW}}\right)$$
(13)$$\Delta G_{h_{0}}^{EL}=\frac{\varepsilon_{r}\varepsilon_{0}\kappa }{2}\left(\zeta_{1}^{2}+\zeta_{3}^{2}\right)\left(1-\coth\left(\kappa h_{0}\right)+\frac{2\zeta_{1}\zeta_{3}}{\zeta_{1}^{2}+\zeta_{3}^{2}}\mathrm{csch}(\kappa h_{0})\right)$$
(14)$$\Delta G_{h_{0}}^{AB}=2\left[\sqrt{\gamma_{2}^{+}}(\sqrt{\gamma_{1}^{-}}+\sqrt{\gamma_{3}^{-}}-\sqrt{\gamma_{2}^{-}})+\sqrt{\gamma_{2}^{-}}(\sqrt{\gamma_{1}^{+}}+\sqrt{\gamma_{3}^{+}}-\sqrt{\gamma_{2}^{+}})-\sqrt{\gamma_{3}^{-}}\sqrt{\gamma_{1}^{+}}-\sqrt{\gamma_{3}^{+}}\sqrt{\gamma_{1}^{-}}\right]$$

The values of $\gamma^{LW}$, $\gamma^{+}$, and $\gamma^{-}$ (mJ∙m^−2^) were determined by solving a Young’s equation group [54]:(15)$$\frac{\left(1+cos\phi \right)}{2}\gamma_{l}^{TOL}=\sqrt{\gamma_{l}^{LW}}\sqrt{\gamma_{s}^{LW}}+\sqrt{\gamma_{l}^{-}}\sqrt{\gamma_{s}^{+}}+\sqrt{\gamma_{l}^{+}}\sqrt{\gamma_{s}^{-}}$$
where the subscripts *l* and *s* denote the probe liquid and solid surface, respectively.

## 3. Results

### 3.1. Impacts of Ca^2+^ Concentration on Filtration Behaviors of HA

Figure 1 shows the membrane filtration flux after different operational steps under different Ca^2+^ concentrations. As displayed in Figure 1, the permeation flux significantly decreases after the filtration of HA suspensions. The flux is only 6.3%, 9.6%, and 18.8% of the virgin membrane for the HA containing Ca^2+^ concentrations of 0, 1.5, and 5.0 mM, respectively. After the cleaning processes of rinsing, deposit removal, and ultrasonic wave treatment, the permeation flux recovers to 41.1%, 59.1%, and 79.5% of the virgin membrane for the HA containing Ca^2+^ concentrations of 0, 1.5, and 5.0 mM, respectively. All in all, the addition of Ca^2+^ leads to a less flux drop as compared with the pure HA. In addition, a higher Ca^2+^ addition corresponds to a higher permeation flux and lower internal blocking resistance (*R_i_*). As shown in Figure 1, the flux decline results from four hydraulic resistances; the effects of Ca^2+^ on HA fouling should be further ascertained and analyzed in each hydraulic resistance.

Figure 2 shows the filtration resistance distribution of HA under different Ca^2+^ concentrations. As displayed in Figure 2, the virgin membrane resistance (*R_m_*) is comparable while the values of the other three filtration resistances (*R_e_*, *R_p_*, and *R_i_*) decrease with the increased Ca^2+^ content. It indicates that the addition of Ca^2+^ can improve the anti-fouling property of HA, and the increase in the Ca^2+^ concentration can enhance this effect. However, unlike the absolute value of the hydraulic resistance, the proportion and contribution of each resistance component in the total hydraulic resistance have different variation trends. The proportion of *R_m_* increases due to the significant reduction of total filtration resistance after the addition of Ca^2+^ into the HA solution. Among the other three resistances, the proportion of *R_i_* is the smallest, and it decreases with the Ca^2+^ concentration (the proportion of *R_i_* is 9.02%, 6.57%, and 4.81% for Ca^2+^ concentrations of 0, 1.5, and 5.0 mM, respectively). Similarly, the ratio of *R_p_* to the total filtration resistance decreases with the increase in the Ca^2+^ content, which is 50.73%, 40.49%, and 32.17%, respectively. On the contrary, the proportion of *R_e_* increases with the Ca^2+^ concentration, which is 33.93%, 43.48%, and 44.32%, respectively. The above results indicate that an increase in Ca^2+^ addition weakens the role of *R_i_* (internal blocking resistance) and *R_p_* (concentration polarization resistance) in the total hydraulic resistance but enhances that of *R_e_* (membrane surface deposit resistance). This result is not completely consistent with previous studies, and further research is required to explore the underlying mechanisms.

### 3.2. Characterization of HA under Different Ca^2+^ Concentrations

#### 3.2.1. FT-IR Spectra Analysis

Figure 3 shows the FTIR spectra of HA samples with different Ca^2+^ concentrations. The broad band around 3300 cm^−1^ represents the O–H stretching vibration of phenolic compounds, and the adsorption peak at 1550 cm^−1^ can be assigned to aromatic C=C stretching and C=O stretching [55,56]. The peak around 1390 cm^−1^ represents the symmetrical stretching vibration of -COO- related to carboxylate. The vibrational frequency in the range of 650–900 cm^−1^ is usually considered aromatic C–H out of plane bending [57]. Obviously, the peak of the HA solution here is stronger than that of other cases, suggesting that the structure of HA has been changed to some extent after the addition of Ca^2+^. However, FTIR is a qualitative characterization method, and the different peak intensities cannot strongly support the different filtration resistances shown in Figure 1 and Figure 2. Therefore, the different filtration performances should be ascribed to other causes.

#### 3.2.2. Particle Size Distribution (PSD) and TOC Removal Measurements

Figure 4 shows the PSD of HA samples containing different Ca^2+^ concentrations. As displayed in Figure 4, the pure HA solution exerts a single peak shape, and the mean size of HA flocs is about 3.31 μm. After adding 1.5 mM Ca^2+^, the floc size of the HA suspension exhibits a double peak shape. The distribution of HA flocs in the ranges of 0–50 μm and 50–500 μm significantly decreases and increases, respectively. As a result, the mean size of HA flocs increases to 76.04 μm. After a further increase in Ca^2+^ concentration to 5.0 mM, the distribution of HA flocs in the ranges of 0–10, 70–105, and 300–500 μm increases, whereas that in the ranges of 10–70 and 105–300 μm decreases. This phenomenon suggests that the increased Ca^2+^ concentration has two different effects on HA. First, the electrostatic shielding effect leads to the compression of some negatively charged HA molecules, which leads to the reduction of some HA flocs. On the other hand, the bridging effect of Ca^2+^ results in the extension of HA molecular chains, which causes an increase in the particle size. Since the mean floc size of HA at the Ca^2+^ concentration of 5.0 mM increases to 107.28 μm, it can be concluded that the bridging effect of Ca^2+^ on HA should be much stronger than the electrostatic effect, and thus results in the formation of larger HA flocs.

The enlarged particle size under the addition of Ca^2+^ is further supported by the optical image (Figure 5) and TOC removal (Figure 6). As shown in Figure 5, after 24 h of natural standing, the HA solutions with different Ca^2+^ concentrations display different sedimentation properties. When the Ca^2+^ concentration is 1.5 mM, the color of the upper layer of the mixed solution is lighter than that of the pure HA solution, although no obvious sedimentation can be seen at the bottom due to the dark color of the HA itself. Nevertheless, when the Ca^2+^ concentration increases to 5.0 mM, it can be seen that there is obvious sedimentation at the bottom of the beaker, and the upper layer solution becomes clearer. High sedimentation corresponds to a larger particle size. The observed phenomena further prove that the size of HA flocs increases with the increased Ca^2+^ concentration. Due to the enhanced sedimentary property, the TOC content in the supernatant before and after filtration significantly drops correspondingly after the addition of Ca^2+^ (Figure 6). In addition, the TOC removal efficiency after filtration also significantly increases from 44.5% (pure HA) to 78.4% (HA + 1.5 mM Ca^2+^) and 74.1% (HA + 5.0 mM Ca^2+^). It indicates that although the addition of Ca^2+^ is beneficial to the removal of TOC in pure HA, the addition of higher concentrations of Ca^2+^ cannot further increase the removal of TOC in the ultrafiltration membrane filtrate. In short, the above results consistently suggest that the bridging effect of Ca^2+^ can increase the particle size of HA. It is widely accepted that a larger particle size generally corresponds to a lower membrane fouling potential [58,59,60], which is well-consistent with the filtration resistance change. Therefore, the floc size change caused by Ca^2+^ addition is considered the main reason for the different filtration resistance of HA.

### 3.3. Thermodynamic Mechanism of HA Fouling Behavior

The adhesion of foulants on the membrane surface is an important process for membrane fouling formation. The XDLVO theory, which has been widely used for the quantitative calculation of the interaction energy between two surfaces, was used to evaluate the adhesion ability of HA with different Ca^2+^ concentrations. The surface contact angle and zeta potential of the PVDF membrane and HA layers with different Ca^2+^ concentrations are listed in Table 1. Based on these data, the interaction energies with separation distance were calculated, and the results are shown in Figure 7. It can be seen that the AB interaction accounts for the vast majority of the total energy for all the scenarios, and thus predominantly manipulates the fouling process. Since the AB interaction energy is positive in all three scenarios, the total interaction energy is always positive regardless of the Ca^2+^ concentration. It indicates that HA particles are difficult to adhere to the membrane surface [61], which well supports the proportion of membrane surface deposit resistance (*R_e_*) in Figure 2 (generally more than 80% while only 33.93–44.32% in this work). In addition, the interaction energy intensity of the HA with Ca^2+^ is always higher than that of the pure HA, suggesting the improved anti-adhesion property of HA by Ca^2+^ addition.

Although the interaction energy between the membrane and HA is repulsive, membrane fouling is unavoidable due to the external drag force. Figure 8 shows the schematic diagram of the hydraulic resistance variation of HA with Ca^2+^ addition in cross-flow filtration. As shown in Figure 8, small-sized HA particles are dramatically reduced due to the Ca^2+^ bridging effect, which leads to a significant reduction in internal blocking resistance (*R_i_*). Moreover, the enlarged floc size not only can prevent the adhesion and accumulation of HA on the membrane surface but also lead to the loose structure of the foulant layer. As a result, concentration polarization resistance (*R_p_*) decreases. Since the absolute value of *R_i_* and *R_p_* decreased more with Ca^2+^, the proportion and contribution of membrane surface deposit resistance (*R_e_*) in the total filtration resistance correspondingly increased.

It should be noted that the result obtained in the current work is not completely consistent with previous studies. The underlying reasons are located in several aspects. The first reason can be attributed to the membrane material. For example, the variation of membrane hydrophilicity and hydrophobicity through membrane modification may lead to dramatic changes in fouling trends. The second reason can be ascribed to the evaluation scope (a single or whole filtration resistance variation). For instance, Lin et al. [34] evaluated the effects of Ca^2+^ on HA fouling only through the interfacial interaction energy change. Chang et al. [36] mainly focused on the HA–Ca^2+^ effects in hydraulically irreversible fouling. The last reason can locate in HA concentration because the effects of specific Ca^2+^ concentration on HA fouling are dependent on the HA concentration. In this study, the SFR of HA decreased with the increase in Ca^2+^ concentration, which is inconsistent with the result (the membrane fouling firstly increased and then decreased with increasing Ca^2+^ concentration) observed by Miao et al. [37]. It is mainly attributed to the different HA concentrations applied. The HA concentration in this study was 100 mg/L, which was one-tenth that of Miao et al. (1 g/L) [37]. The Ca^2+^ concentrations selected in the current work exceeded the critical concentration, and, thus, only a monotonically decreasing variation trend was observed. As HA and Ca^2+^ concentration in the natural water bodies is in the range of 0.02−30 mg/L and 0.5–3 mM, respectively [26], the coexistence of HA and Ca^2+^ in natural water and wastewater generally can achieve a membrane mitigation effect.

## 4. Conclusions

In the current work, the effects of Ca^2+^ on HA fouling were evaluated with the Darcy–Poiseuille model through four different hydraulic resistance components. The results show that the increase in Ca^2+^ concentration improved the filtration flux and reduced the absolute value of each hydraulic resistance. Unlike the absolute value of the hydraulic resistance, the proportion and contribution of each resistance component in the total hydraulic resistance have different variation trends with the Ca^2+^ concentration. An increase in Ca^2+^ addition (0 to 5.0 mM) weakened the role of internal blocking resistance (*R_i_*, 9.02% to 4.81%) and concentration polarization resistance (*R_p_*, 50.73% to 32.17%) in the total hydraulic resistance but enhanced that of the membrane surface deposit resistance (*R_e_*, 33.93% to 44.32%). A series of characterizations consistently suggest that the enlarged particle size caused by Ca^2+^ addition was the main reason for the different filtration resistance of HA. The calculation results with the XDLVO theory further reveal that the anti-adhesion property of HA was improved due to the bridging effect of Ca^2+^. This work revealed the impacts of Ca^2+^ on HA fouling in the PVDF ultrafiltration membrane and the underlying causes and demonstrated the feasibility to mitigate HA fouling in the PVDF ultrafiltration membrane by adding Ca^2+^.

## Figures and Tables

**Figure 1 membranes-12-01033-f001:**
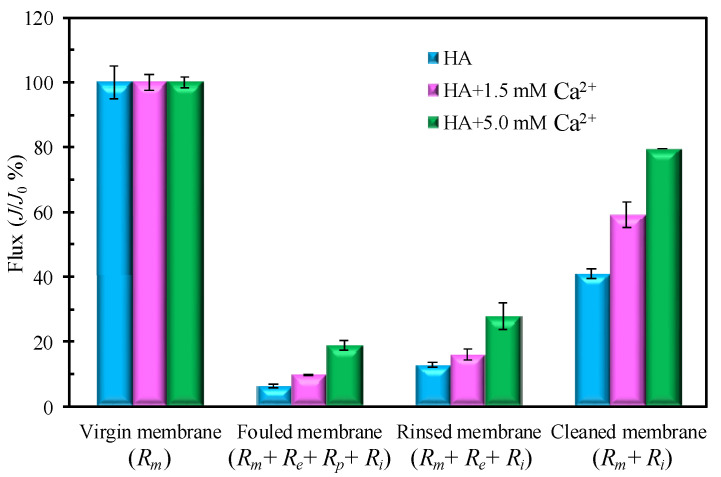
Comparison of membrane filtration flux after different operational steps under different Ca^2+^ concentrations (∆*P* = 2 bar).

**Figure 2 membranes-12-01033-f002:**
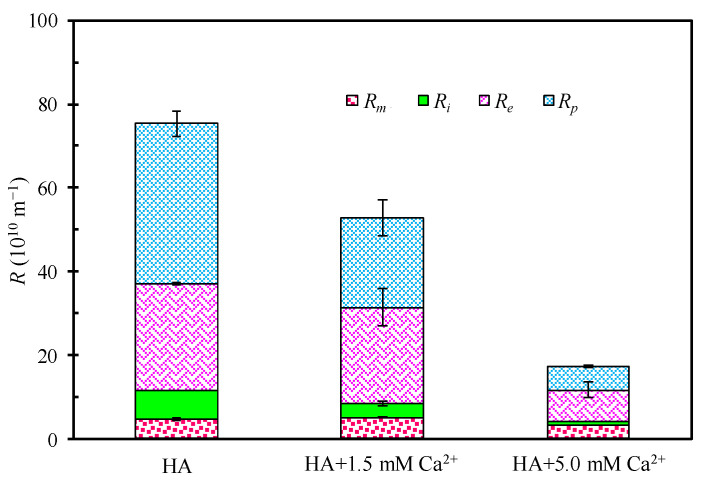
Influence of Ca^2+^ concentration on experimental hydraulic resistances (*R_p_*, *R_e_*, *R_i_*, *R_m_*) of HA (∆*P* = 2 bar).

**Figure 3 membranes-12-01033-f003:**
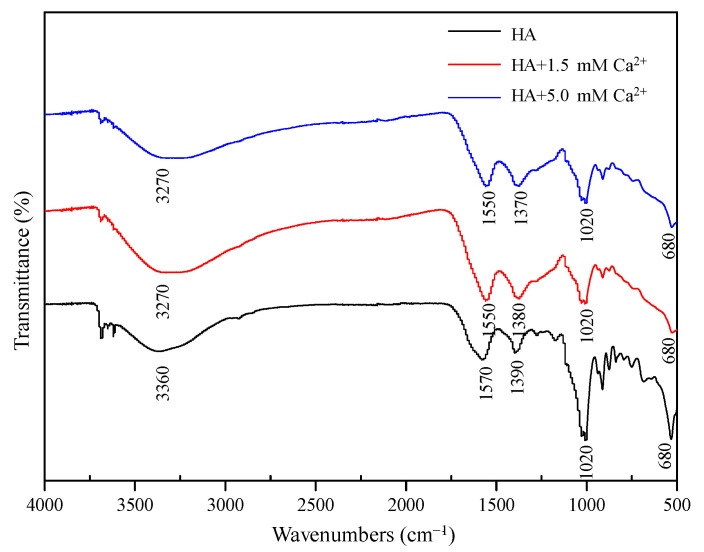
FTIR spectra of HA samples with different Ca^2+^ concentrations.

**Figure 4 membranes-12-01033-f004:**
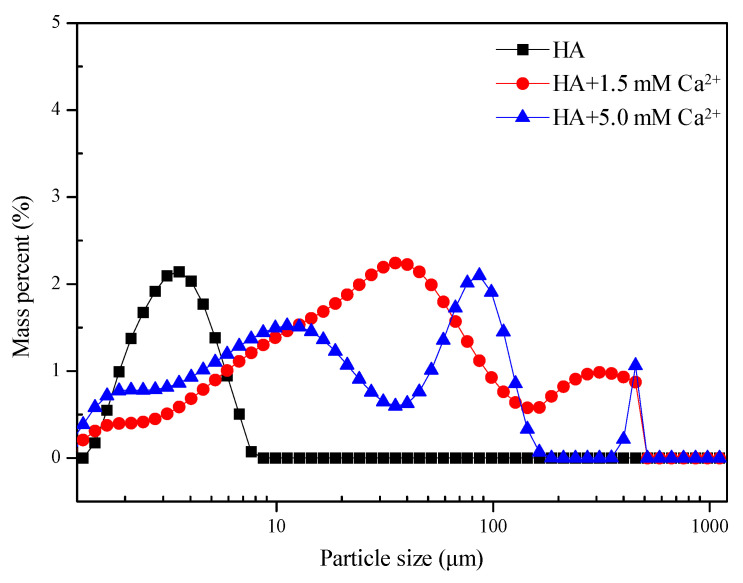
PSD of HA solution with different Ca^2+^ concentrations.

**Figure 5 membranes-12-01033-f005:**
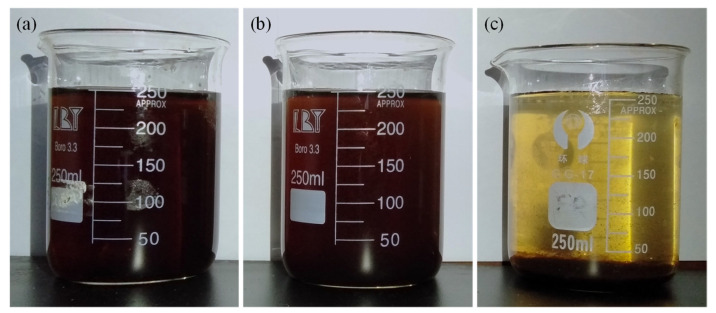
Optical images of HA solutions prepared after 24 h natural sedimentation with Ca^2+^ concentration of (**a**) 0 mM, (**b**) 1.5 mM, and (**c**) 5.0 mM (HA concentration = 100 mg/L).

**Figure 6 membranes-12-01033-f006:**
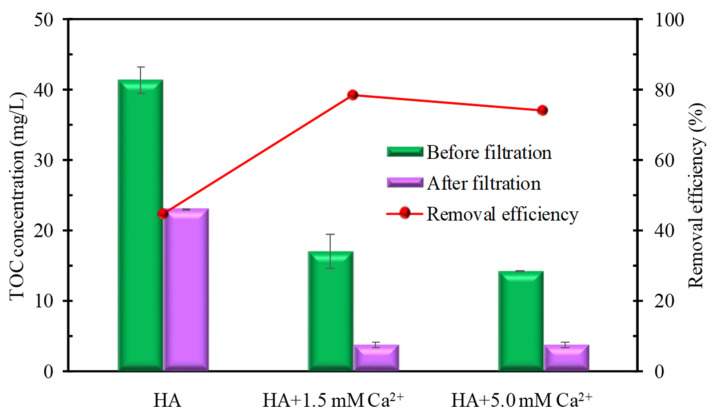
TOC content and removal efficiency of HA with different Ca^2+^ concentrations.

**Figure 7 membranes-12-01033-f007:**
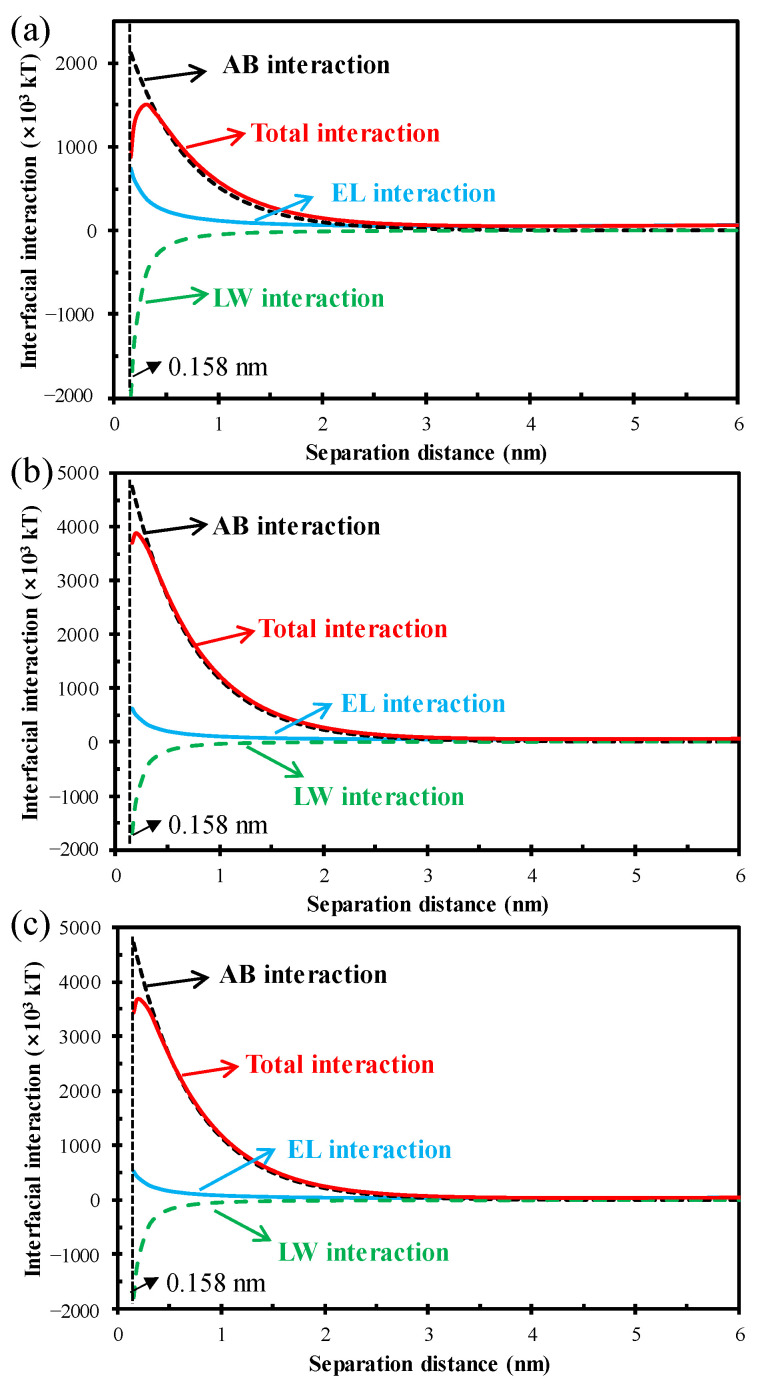
Profiles of interfacial interaction energies between PVDF membrane and HA with different Ca^2+^ concentrations (**a**) 0 mM, (**b**) 1.5 mM, and (**c**) 5.0 mM.

**Figure 8 membranes-12-01033-f008:**
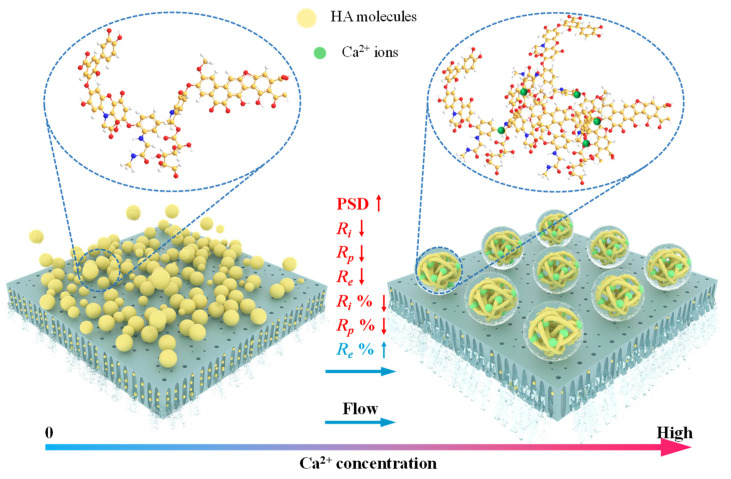
Schematic diagram of hydraulic resistance variation of HA with Ca^2+^ addition in cross-flow filtration.

**Table 1 membranes-12-01033-t001:** Surface characteristics in terms of contact angle of three probe liquids and zeta potential of the PVDF membrane and HA solutions with different Ca^2+^ concentration.

Materials	Contact Angle (°)			Zeta Potential (mV)
	Water	Glycerol	Diiodomethane	
PVDF membrane	62.16 ± 0.10	57.22 ± 1.47	23.15 ± 0.82	−25.21 ± 2.46
HA	48.93 ± 0.28	70.99 ± 0.99	34.36 ± 0.51	−26.67 ± 0.50
HA + 1.5 mM Ca^2+^	42.36 ± 0.47	74.25 ± 0.18	41.36 ± 0.10	−22.87 ± 0.50
HA + 5.0 mM Ca^2+^	41.75 ± 0.12	73.64 ± 0.31	38.61 ± 0.08	−19.03 ± 0.60

## Data Availability

The data presented in this study are available on request from the corresponding author.

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
