# Peer review of "Impacts of Calcium Addition on Humic Acid Fouling and the Related Mechanism in Ultrafiltration Process for Water Treatment"

_membranes, 2022, doi:10.3390/membranes12111033_

Round 1

Reviewer 1 Report

The authors presented a paper entitled “Impacts of calcium addition on humic acid fouling and the related mechanism in ultrafiltration process for water treatment”. Bellow, some comments which should be reported.

2.2. Filtration resistance tests

Line 101: Please specify UF equipment.

Line 104: should be 25 m2

Line 113: What the authors mean: The extract The membrane filtration resistances…

2.3. Analytical methods

Please check the format for the equipment required. 

How do the authors measure Ca2+? It is not mentioned in the analytical methods.

3.1. Impacts of Ca2+ concentration on the filtration behaviors of HA

How do the pH and electrical conductivity change during the process in the solutions?

I think this result is not so clear and the authors can’t find a relation between previous works and they mention it.

Line 273: Table 1.

Please add error bars for all the values.

References.

Please, use the MDPI format to refer articles.  

Reviewer 2 Report

Zou et al. used PVDF membrane for studying the impact of calcium ions on humic acid fouling in ultrafiltration process. The role of calcium in UF membrane fouling, mediated by humic acid, has been well studied in the literatures, even for PVDF based membrane Journal of Membrane Science 545 (2018): 81-87. So, I do not think the study brings new insights and has no novelty to publish in Membrane.   

Reviewer 3 Report

This study investigated the impact of Ca2+ on the HA fouling of ultrafiltration membranes and provided informative information about the impacts of Ca2+ on HA fouling. Overall, the research is designed appropriately, and methods are adequately described. I think this paper could be accepted after minor revision.

Here are some of my comments, and hope these comments could improve this paper’s quality.

1.      Line 80-81: Please add citations to support your statement “Actually, PVDF is one of the most widely used membrane materials in wastewater treatment.”

2.      Line 97-98: please justify the use of 100 mg/L humic acids in your experiments. Is this concentration reported widely in actual environments?

3.      Please check the manuscript and fix the error” Error! Reference source not found. ”

4.      What is the normal Ca concentration in wastewater? Please justify Ca (e.g., 1.5 mM and 5 mM) concentration used in your experiments.

5.      Why there are no error bars in figure 6?

Round 2

Reviewer 1 Report

The article can be accepted in its present form.